# Cost-effectiveness and budget impact analyses of dengue vaccination in Indonesia

**Auliya Abdurrohim Suwantika** [1,2,3]*, **Woro Supadmi** [1,4], **Mohammad Ali**[5], **Rizky Abdulah** [1,2]

**1** Department of Pharmacology and Clinical Pharmacy, Faculty of Pharmacy, Universitas Padjadjaran, Bandung, Indonesia, **2** Center of Excellence in Higher Education for Pharmaceutical Care Innovation, Universitas Padjadjaran, Bandung, Indonesia, **3** Center for Health Technology Assessment, Universitas Padjadjaran, Bandung, Indonesia, **4** Faculty of Pharmacy, Universitas Ahmad Dahlan, Yogyakarta, Indonesia, **5** Faculty of Educational Sciences, Universitas Pendidikan Indonesia, Bandung, Indonesia

* auliya@unpad.ac.id

## Abstract

Despite the fact that the incidence and mortality rates due to dengue virus (DENV) infection in Indonesia are relatively high, dengue vaccination has not yet been introduced. This study aimed to analyse the cost-effectiveness and the budget impact of dengue vaccination in Indonesia by taking the potential of pre-vaccination screening into account. An age-structured decision tree model was developed to assess the cost-effectiveness value by applying a single cohort of 4,710,100 children that was followed-up in a 10-year time horizon within a 1-year analytical cycle. The budget impact was analysed in a 5-year period (2020–2024) by considering provinces' readiness to introduce dengue vaccine and their incidence rate of DENV infection in the last 10 years. Vaccination that was coupled with pre-vaccination screening would reduce dengue fever (DF), dengue haemorrhagic fever (DHF) and dengue shock syndrome (DSS) by 188,142, 148,089 and 426 cases, respectively. It would save treatment cost at $23,433,695 and $14,091,642 from the healthcare and payer perspective, respectively. The incremental cost-effectiveness ratios (ICERs) would be $5,733 and $5,791 per quality-adjusted-life-year (QALY) gained from both perspectives. The most influential parameters affecting the ICERs were probability of DENV infection, vaccine efficacy, under-reporting factor, vaccine price, case fatality rate and screening cost. It can be concluded that dengue vaccination and pre-vaccination screening would be cost-effective to be implemented in Indonesia. Nevertheless, it seems unaffordable to be implemented since the total required cost for the nationwide vaccination would be 94.44% of routine immunization budget.

## Author summary

Up to now, dengue vaccination has not yet been included into the national immunization program in Indonesia. An age-structured decision tree model was developed in this study to assess the cost-effectiveness and the budget impact of dengue vaccination in Indonesia in 2020–2024, which was based on country specific data. The result confirmed that vaccination and pre-vaccination screening programs would be cost-effective to be implemented in

**Data Availability Statement:** All relevant data are within the manuscript and its Supporting Information files.

**Funding:** This study was funded by the Ministry of Research, Technology and Higher Education,

Republic of Indonesia (Grant number: 391/UN6.O/
LT/2018) (AAS). The funder had no role in study
design, data collection and analysis, decision to
publish, or preparation of the manuscript.

**Competing interests:** The authors have declared
that no competing interests exist.

Indonesia. Nevertheless, it seems unaffordable to be implemented since the total required
cost for the nationwide vaccination would be 94.44% of routine immunization budget.

## Introduction

Dengue, the most health threatening vector-borne viral disease in the world, was associated with
approximately 3.2 million cases in 2015, as reported by the World Health Organization (WHO)
[1]. In particular, more than 70% or about 1.8 billion populations in Asia-Pacific region contrib-
ute the most to the overall burden of dengue in the world [2]. A relatively high incidence of den-
gue virus (DENV) infection in this region is mainly caused by secondary infection, high level of
endemicity and all four DENV serotypes (DENV-1, DENV-2, DENV-3 and DENV-4) continu-
ally co-circulate [3–5]. This situation also happens in Indonesia, a country with the second high-
est incidence rate of DENV infection in the world after Brazil [1]. *Ae. aegypti* and *Ae. albopictus*
are the primary and secondary vectors for dengue transmission almost in all provinces, respec-
tively [6], and all four DENV serotypes circulated most years [7]. The manifestations of DENV
infection range from asymptomatic infection or dengue fever (DF), to the more life-threatening
forms, dengue haemorrhagic fever (DHF) and dengue shock syndrome (DSS) [7].

As a disease of great importance for public health in Southeast Asia, dengue is responsible
with the annual economic burden in 12 countries at $950 million [8]. The annual cost in Indone-
sia was estimated to be $323 million, according to a study by Shepard *et al.* in 2013 [8]. Further-
more, another study by Nadjib *et al.* estimated the annual economic burden of dengue in
Indonesia would be $381.15 million for hospitalization ($355.2 million) and outpatient cases
($26.2 million) [9]. To deal with the epidemiological and economic burden of dengue, the WHO
targeted to reduce dengue mortality by 50% at the end of 2020 [2]. Vaccinations to prevent infec-
tious diseases have been estimated to be cost-effective strategies since 1993 [10]. For dengue vacci-
nation, it has been expected to be a highly cost-effective or even cost-saving intervention in ten
endemic countries [11]. On the other hand, a recent study confirmed an elevated of severe disease
in vaccinees with no prior DENV exposure, which made an end to the implementation of dengue
vaccine in the Philippines [12]. Therefore, the World Health Organization (WHO) recom-
mended countries to integrate a pre-vaccination screening in dengue vaccination program [1, 2].

Up to now, there is only one available dengue vaccine (CYD-TDV) that has been marketed
in Indonesia since 2016 [11]. Despite the fact that the incidence and mortality rates due to
DENV infection in Indonesia are relatively high [1, 9], dengue vaccination has not yet been
included into the national immunization program (NIP). This situation might be caused by
the lack of economic evaluation studies, which was based on country specific data. In particu-
lar, the recommendation of pre-vaccination screening might affect the favourable cost-effec-
tiveness value of dengue vaccination to be implemented in Indonesia. Hence, this study aimed
to analyse the cost-effectiveness and the budget impact of dengue vaccination in Indonesia by
taking the potential of pre-vaccination screening into account.

## Methods

### Ethics statement

The ethical permission was obtained from the Ethics Committee of Universitas Padjadjaran,
Indonesia (approval number: 65/UN6.C10/PN/2017) and this study was conducted in accor-
dance with the Declaration of Helsinki.

## Model

An age-structured decision tree model was developed to assess the cost-effectiveness and the budget impact of dengue vaccination in Indonesia in 2020–2024. To estimate the cost-effectiveness value, a single cohort of 4,710,100 children [13], started at the age of 9 years old (recommended age for dengue vaccination in Indonesia), was followed-up in a 10-year time horizon within a 1-year analytical cycle (see S1 Appendix). The budget impact was analysed in a 5-year period (2020–2024) by considering provinces' readiness to introduce dengue vaccine and their incidence rate of DENV infection in the last 10 years. In 2020, dengue vaccine was targeted to be introduced in 6 provinces (Bali, Kalimantan Timur, Jakarta, Kalimantan Utara, Kepulauan Riau and Yogyakarta). In 2021, 6 more provinces (Sulawesi Tengah, Kalimantan Barat, Kalimantan Tengah, Sumatera Barat, Sulawesi Utara and Jawa Barat) would be added. In 2022, the introduction program would be expanded in 7 provinces (Aceh, Bengkulu, Sumatera Utara, Sulawesi Tenggara, Kalimantan Selatan, Sulawesi Selatan and Bangka Belitung). In 2023, 7 more provinces (Lampung, Jawa Timur, Jawa Tengah, Riau, Banten, Sulawesi Barat and Gorontalo) would be included. At the end, nationwide vaccination was targeted to be implemented in 2024. The model was programmed in Microsoft Excel and @Risk was used for probabilistic sensitivity analysis (see Fig 1) [14].

## Disease burden estimates

To estimate the epidemiological burden of dengue in Indonesia, we applied probabilities of primary (0.1531, 0.2306 and 0.0015 for DF, DHF and DSS, respectively) and secondary

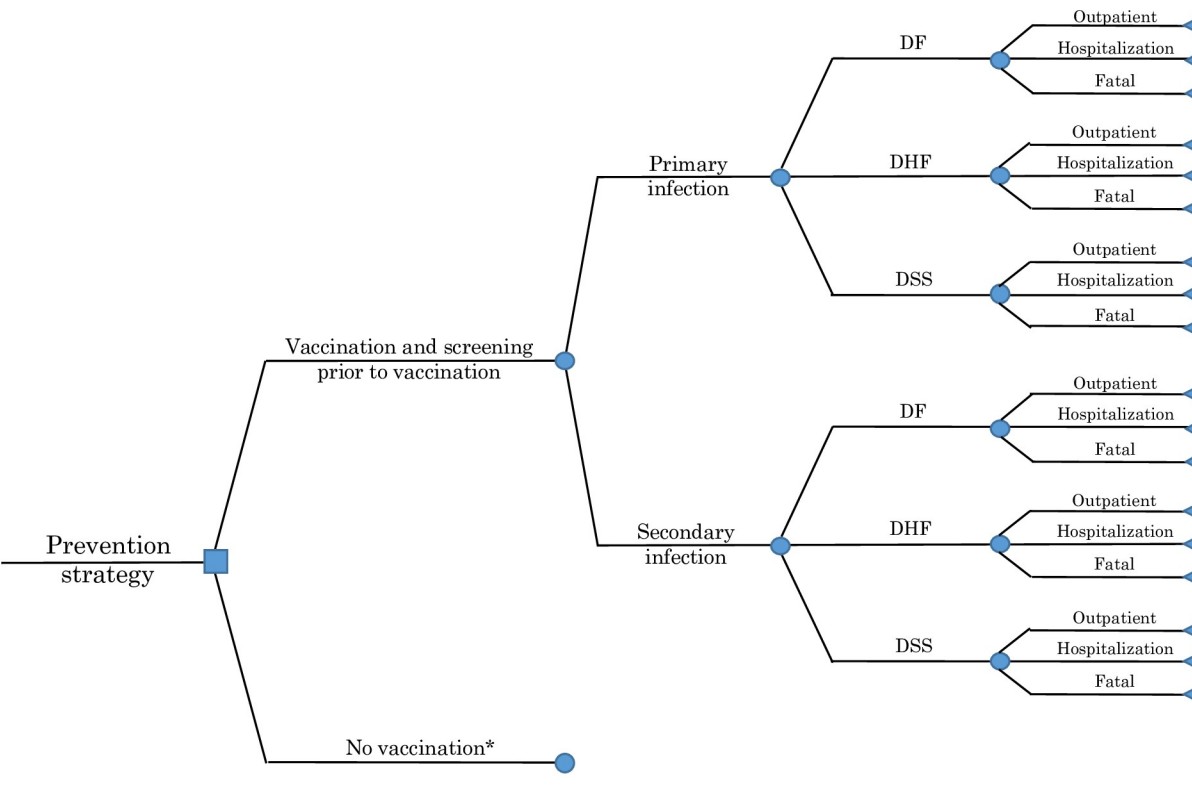

*same branches as baseline are applied

**Fig 1. Decision tree model.**

infection (0.2449, 0.3674 and 0.0025 for DF, DHF and DSS, respectively), outpatient (0.6800, 0.2610 and 0.0000 for DF, DHF and DSS, respectively) and hospitalization cases (0.3200, 0.7390 and 1.0000 for DF, DHF and DSS, respectively), according to the national administrative data and the results of several previous studies [7, 9, 15–17]. For DF outpatient cases, we estimated the probabilities of primary and secondary infections would be 0.1041 (0.1531*0.6800) and 0.1655 (0.2449*0.6800), respectively. For DF hospitalization cases, we estimated the probabilities of primary and secondary infections would be 0.0490 (0.1531*0.3200) and 0.0784 (0.2449*0.3200), respectively. We also estimated the probabilities of primary and secondary infections for DHF outpatient cases would be 0.0602 (0.2306*0.2610) and 0.0959 (0.3674*0.2610), respectively. For DHF hospitalization cases, the probabilities of primary and secondary infections would be 0.1704 (0.2306*0.7390) and 0.2715 (0.3674*0.7390), respectively. In contrast with other cases, we estimated there would not be primary and secondary infections for DSS outpatient cases. For DSS hospitalization cases, we estimated the probabilities of primary and secondary infections would be 0.0015 (0.0015*1.0000) and 0.0025 (0.0025*1.0000), respectively. In particular, we applied case fatality rate at 0.83% by considering the average rate in the last 10 years in Indonesia [15]. Since the number of under-reporting cases due to DENV infection in Indonesia is very high, adjustment factors for outpatient and inpatient cases were applied at 45.90 and 7.65, respectively [9]. More detailed information can be seen in S2 Appendix.

## Pre-vaccination screening

We applied a strategy of routine vaccination in which a proportion of children underwent serological screening and vaccination in the event of a positive result would be implemented on their ninth birthday [18]. As its consequence, the intervention coverage (*i.e.*, the proportion of children screened) would represent an upper limit on the proportion of vaccine-eligible children [18]. We did not consider the potential impact of vaccination on individuals with no prior dengue exposure since vaccination could increase the risk of severe dengue in those who have not previously been infected [19]. According to a previous study on assessing the impact of dengue vaccination following screening for prior exposure [18], seropositivity among 9 years old population ($SP_9$) could be defined as:

$$SP_9 = \text{prior exposure in 9 years old before vaccination } (PE_9) \text{ x sensitivity} + (1 - PE_9) \text{ x } (1 - \text{specificity}).$$

We applied $PE_9$ in Indonesia would be 83.1% (95% CI: 77.1–89.0%), according to a previous study by Prayitno *et al.* [20]. In addition, we applied a sensitivity of 95.2% (95%CI: 94.2–96.2%) and a specificity of 93.4% (95%CI: 89.6–97.2%) for determining dengue virus serostatus by indirect IgG ELISA with false positivity and negativity of 6.6% and 4.8%, respectively [21]. According to these data, we calculated $SP_9$ in Indonesia would be 80.23% [20, 21].

## Vaccine characteristics

A vaccine efficacy of 44% was applied from a meta-analysis by using the random-effects model, which estimated the vaccine efficacy with a range from 25% to 59% and high heterogeneity of 80.1% from 7 clinical trials that were included [22]. Currently, CYD-TDV should be used within the indicated age range, which is started at the age of 9 years old. As a 3-dose vaccine, it should be given 6 months apart [19]. The coverage of dengue vaccination was defined as:

$$\text{Dengue vaccination coverage} = \text{targeted coverage x } SP_9.$$

Considering the value of $SP_9$ at 80.23% and the coverage of basic childhood immunization at 87.56% [15], we estimated the coverage of dengue vaccination would be 70.25%.

### Treatment and vaccination costs

Cost analyses in this study were conducted from two perspectives: healthcare (direct medical costs) and payer perspective (all costs covered by the Indonesian National Health Insurance System/BPJS Kesehatan). Healthcare costs of outpatient and inpatient cases were derived from a study on economic burden of dengue in Indonesia [9]. Payer costs of outpatient and inpatient cases were derived from the tariff of capitation and Indonesia case-based groups (INA-CBGs), respectively [23, 24].

A vaccine price per dose of $20 and pre-vaccination screening cost of $10 were applied from a study by Zeng *et al.*, which focused on the cost-effectiveness of dengue vaccination in 10 endemic countries, including Indonesia [11]. Cost of vaccine administration ($3.42) and wastage (10%) were also derived from the same study [11]. Since dengue vaccine was reported to have common minor side effects (*e.g.*, localized pain, swelling, fever and aches), we considered a side effect cost at $0.31 according to a previous study in Thailand [25]. All cost items from different currencies and years were converted into 2018 US$ by using purchasing power parity (PPP) [26]. In particular, all costs were discounted with an annual rate of 3% (see Table 1 and S3, S4 and S5 Appendices).

### Outcome measures

Applying a questionnaire of EuroQol 5 dimensions 5 levels [27], we calculated quality-adjusted-life-year (QALY) losses in Indonesia due to DENV infection by delivering a retrospective pre-post questionnaire to 144 patients in 3 cities (Jakarta, Bandung and Yogyakarta), which represented regions with high prevalence of DENV infection in Indonesia. We estimated QALY losses in outpatient, hospitalization and fatal cases would be 0.00004, 0.00018 and 1, respectively. All outcome measures were discounted at a 3% rate (see Table 1 and S6 Appendix).

### Cost-effectiveness and budget impact analyses

The incremental cost-effectiveness ratio (ICER) was evaluated by using the Commission for Macroeconomics and Health (CMH) thresholds, which were based on the human capital theory and the argument of saving a life year could create market income at least equivalent to the average wage [28, 29]. Interventions with ICER below GDP per capita are highly cost-effective, while those between 1–3 times GDP per capita are cost-effective, and those above 3 times GDP per capita are cost-ineffective [30, 31]. Univariate sensitivity analysis was performed to investigate the effects of different input parameters on cost and health outcomes, by mostly varying each parameter at value of ± 25% while keeping other parameters constant. In addition, probabilistic sensitivity analysis was also performed by running 5,000 Monte Carlo simulations.

### Results

Applying a cohort of 4,701,100 children [13], nationwide vaccination would reduce DF, DHF and DSS by 188,142, 148,089 and 426 cases, respectively. In particular, vaccination would reduce DF by 173,023, 13,570 and 1,549 for outpatient, hospitalization and fatal cases, respectively. It would reduce DHF by 99,782, 47,088 and 1,219 for all cases, respectively. It also would reduce DSS by 0, 426 and 0 for all cases, respectively (see Fig 2). Additionally,

**Table 1. Input parameters.**

| Parameters | Value | Distribution | Reference |
|---|---|---|---|
| *Epidemiology* | | | |
| Probability of primary infection (DF outpatient) | 0.000048 | Dirichlet | [7, 9, 15–17] |
| Probability of primary infection (DF hospitalization) | 0.000023 | Dirichlet | [7, 9, 15–17] |
| Probability of secondary infection (DF outpatient) | 0.000077 | Dirichlet | [7, 9, 15–17] |
| Probability of secondary infection (DF hospitalization) | 0.000036 | Dirichlet | [7, 9, 15–17] |
| Probability of primary infection (DHF outpatient) | 0.000028 | Dirichlet | [7, 9, 15–17] |
| Probability of primary infection (DHF hospitalization) | 0.000079 | Dirichlet | [7, 9, 15–17] |
| Probability of secondary infection (DHF outpatient) | 0.000044 | Dirichlet | [7, 9, 15–17] |
| Probability of secondary infection (DHF hospitalization) | 0.000125 | Dirichlet | [7, 9, 15–17] |
| Probability of primary infection (DSS outpatient) | 0.000000 | Dirichlet | [7, 9, 15–17] |
| Probability of primary infection (DSS hospitalization) | 0.000001 | Dirichlet | [7, 9, 15–17] |
| Probability of secondary infection (DSS outpatient) | 0.000000 | Dirichlet | [7, 9, 15–17] |
| Probability of secondary infection (DSS hospitalization) | 0.000001 | Dirichlet | [7, 9, 15–17] |
| Case fatality rate | 0.83% | Dirichlet | [15] |
| Under reporting factor for outpatient | 45.90 | Dirichlet | [9] |
| Under reporting factor for hospitalization | 7.65 | Dirichlet | [9] |
| *Costs* | | | |
| Healthcare cost of outpatient | $19.22 | Gamma | [9] |
| Healthcare cost of hospitalization | $297.79 | Gamma | [9] |
| Payer cost of outpatient | $0.62 | Gamma | [24] |
| Payer cost of hospitalization | $227.94 | Gamma | [23] |
| Vaccine price per dose | $20.00 | Alternative scenario | [11] |
| Cost of vaccine administration | $3.42 | Alternative scenario | [11] |
| Screening cost | $10.00 | Alternative scenario | [11] |
| Side effect | $0.31 | Alternative scenario | [25] |
| Wastage (10%) | $2.00 | Alternative scenario | [11] |
| *Vaccine characteristics* | | | |
| Vaccine efficacy | 44.00% | Alternative scenario | [22] |
| Basic immunization coverage | 87.56% | Alternative scenario | [15] |
| Dengue vaccination coverage | 70.25% | Alternative scenario | Calculation |
| Schedule interval (3-dose for >9 years old) | 6-month | | [19] |
| *Dengue screening* | | | |
| Seropositivity among 9 years old | 80.23% | Alternative scenario | [20, 21] |
| Prior exposure in 9 years old before vaccination | 83.10% | Alternative scenario | [20] |
| Sensitivity | 95.20% | Alternative scenario | [21] |
| Specificity | 93.40% | Alternative scenario | [21] |
| *Utilities* | | | |
| QALYs lost of outpatient | 0.00004 | Alternative scenario | Calculation |
| QALYs lost of hospitalization | 0.00018 | Alternative scenario | Calculation |
| QALYs lost of fatal | 1.00000 | Alternative scenario | Calculation |
| *Others* | | | |
| Targeted population (nationwide) | 4,701,100 | | [13] |
| Discount rate | 3.00% | | [30] |
| Time horizon | 10 years | | [11] |

vaccination would save treatment cost at $23,433,695 and $14,091,642 from the healthcare and payer perspective, respectively (see Fig 3).

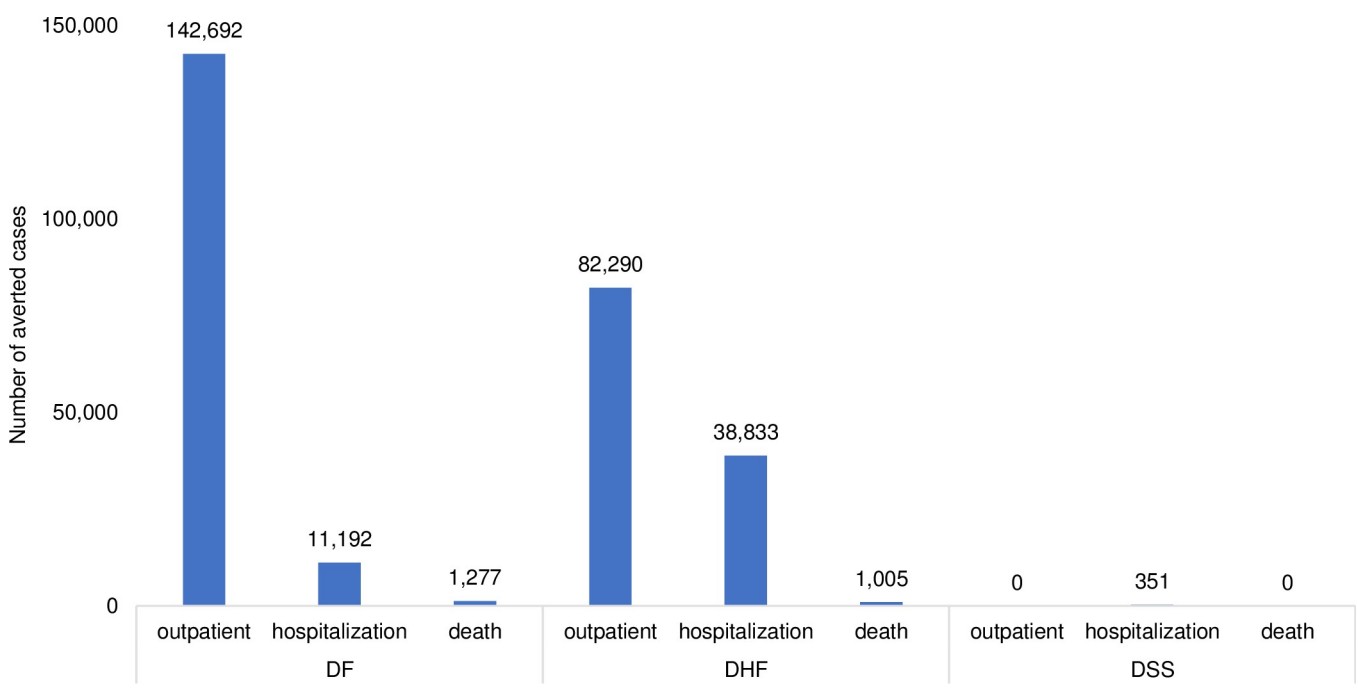

**Fig 2. Number of averted cases due to vaccination and screening prior to vaccination (2020–2024).**

An introduction scenario of dengue vaccination was developed by prioritizing provinces with high incidence rate. We targeted number of eligible populations to be vaccinated would be 439,400; 1,681,880; 2,443,740; 4,123,700 and 4,701,100 in 2020–2024. We estimated the

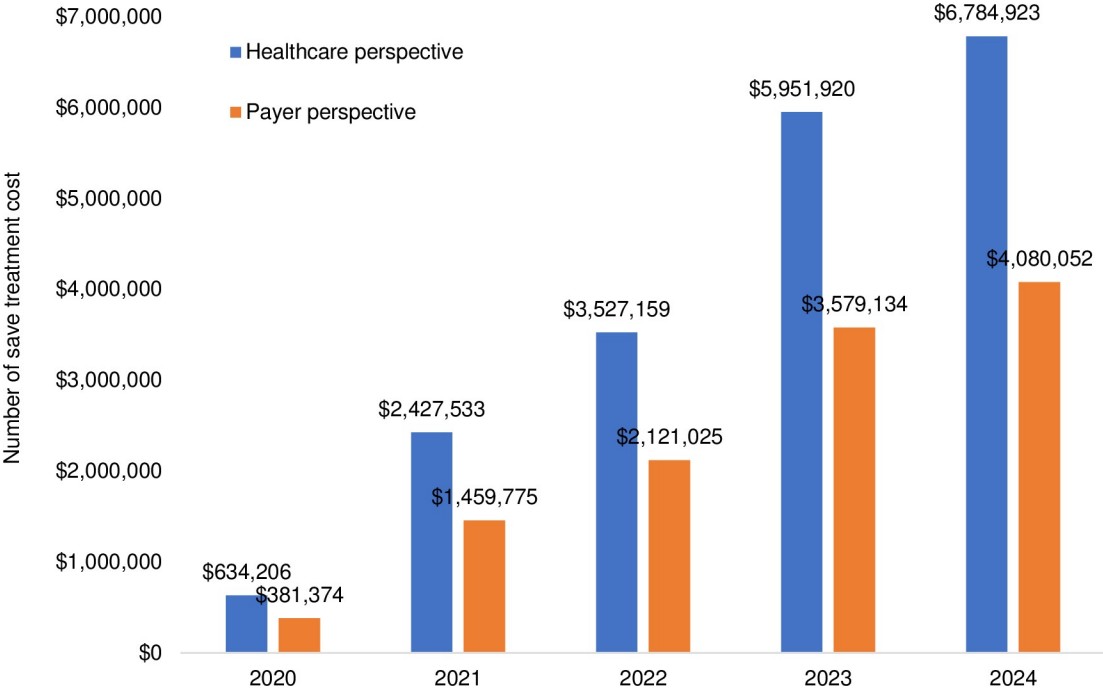

**Fig 3. Number of saved treatment cost due to vaccination and screening prior to vaccination (2020–2024).**

introduction cost (*e.g.*, social mobilization, microplanning, training and supervision-monitoring) by considering unit cost per activity in each district, which was based on the result of in-depth interview with respondents from the Ministry of Health. Learning from the experience on the introduction of HPV vaccine, the average cost for social mobilization, microplanning, training and supervision-monitoring would be $3,106, $243, $5,048 and $512, respectively. Hence, the introduction cost per child for dengue vaccination was estimated to be $0.97. Applying a vaccine price per dose of $20, pre-vaccination screening cost of $10, vaccination coverage of 70.25% and wastage rate of 10%, we estimated total vaccination cost would be approximately $25.5 million, $96.9 million, $140.6 million, $236.4 and $335.5 in 2020–2024. The vaccination cost (3-dose vaccine) per child was estimated to be $71.14. Considering the introduction cost, total required cost per child would be $72.11. The cost-effectiveness values of dengue vaccination were estimated to be $335,696 and $339,071 per death averted from the healthcare and payer perspective, respectively. In the context of cost per QALY-gained, the incremental cost-effectiveness ratios (ICERs) would be $5,733 and $5,791 per QALY-gained from the healthcare and payer perspective, respectively (see S7 Appendix). Considering the GDP per capita in Indonesia of $3,859 [32], the results confirmed that the dengue vaccination would be cost-effective from both perspectives since the ICERs were between 1–3 times GDP per capita. Next to the ICERs, we compared the required cost for dengue vaccination with total national healthcare budget and routine immunization budget. Compared with total national healthcare budget, the required cost for dengue vaccination would be 0.57%, 2.12%, 3.04%, 5.03% and 7.04% in 2020–2024. Compared with routine immunization budget, it would be 8.70%, 31.56%, 43.47%, 69.71% and 94.44% in the same period (see Table 2).

The effects of varying input parameters on the ICERs are shown in a tornado chart (see Fig 4). From the healthcare perspective, the result confirmed that vaccine efficacy, under-reporting factor for outpatient, probability of secondary infection (DF outpatient), vaccine price per dose, case fatality rate, probability of primary infection (DF outpatient), probability of secondary infection (DHF outpatient), screening cost, probability of primary infection (DHF outpatient), probability of secondary infection (DHF hospitalization), under-reporting factor for hospitalization and probability of primary infection (DHF hospitalization) are the most influential parameters affecting cost-effectiveness value.

Applying a threshold ICER of $3,859 (GDP per capita), the probability for the vaccination program to be cost-effective would be 0% from both perspectives. Applying a threshold ICER of $5,733 (ICER from the healthcare perspective), the probability for the vaccination program to be cost-effective would be 50.5% and 23.1% from the healthcare and payer perspective, respectively (see Fig 5). The affordability related to the required budget of programs from both perspectives are shown in cost-effectiveness affordability curves. Dengue vaccination with the vaccine price of $20 per dose would be implementable when the budget exceeds $382.37 million and $375.36 million from the healthcare and payer perspective, respectively (see Fig 6).

## Discussion

Vaccination has been proven to be one of the most significant interventions in reducing vaccine-preventable diseases. However, the introduction of new vaccines tends to be delayed in countries with limited immunization budget, such as Indonesia, due to the lack of cost–effectiveness studies, inadequate health systems, financial barriers and insufficient concern from the government [33–36]. This study confirmed that a nationwide dengue vaccination appears to be one of promising interventions to prevent DENV infection by showing potential benefits on reducing DF, DHF and DSS by 188,142, 148,089 and 426 cases, respectively. Furthermore, a nationwide vaccination and screening prior to vaccination would yield ICERs at $5,733 and

**Table 2. Introduction scenario of dengue vaccination in Indonesia.**

| Year | 2020 | 2021 | 2022 | 2023 | 2024 |
|---|---|---|---|---|---|
| **Province** | Bali | Bali | Bali | Bali | All provinces (nationwide) |
| | Kalimantan Timur | Kalimantan Timur | Kalimantan Timur | Kalimantan Timur | |
| | Jakarta | Jakarta | Jakarta | Jakarta | |
| | Kalimantan Utara | Kalimantan Utara | Kalimantan Utara | Kalimantan Utara | |
| | Kepulauan Riau | Kepulauan Riau | Kepulauan Riau | Kepulauan Riau | |
| | Yogyakarta | Yogyakarta | Yogyakarta | Yogyakarta | |
| | | Sulawesi Tengah | Sulawesi Tengah | Sulawesi Tengah | |
| | | Kalimantan Barat | Kalimantan Barat | Kalimantan Barat | |
| | | Kalimantan Tengah | Kalimantan Tengah | Kalimantan Tengah | |
| | | Sumatera Barat | Sumatera Barat | Sumatera Barat | |
| | | Sulawesi Utara | Sulawesi Utara | Sulawesi Utara | |
| | | Jawa Barat | Jawa Barat | Jawa Barat | |
| | | | Aceh | Aceh | |
| | | | Bengkulu | Bengkulu | |
| | | | Sumatera Utara | Sumatera Utara | |
| | | | Sulawesi Tenggara | Sulawesi Tenggara | |
| | | | Kalimantan Selatan | Kalimantan Selatan | |
| | | | Sulawesi Selatan | Sulawesi Selatan | |
| | | | Bangka Belitung | Bangka Belitung | |
| | | | | Lampung | |
| | | | | Jawa Timur | |
| | | | | Jawa Tengah | |
| | | | | Riau | |
| | | | | Banten | |
| | | | | Sulawesi Barat | |
| | | | | Gorontalo | |
| **Total provinces** | 6 | 12 | 19 | 26 | 34 |
| **Total districts** | 42 | 144 | 271 | 392 | 514 |
| **Total targeted population** | 439,400 | 1,681,880 | 2,443,740 | 4,123,700 | 4,701,100 |
| **Total introduction cost** | $374,172 | $908,703 | $1,131,424 | $1,077,971 | $1,086,880 |
| **Introduction cost per child** | | | | | $0.97 |
| **Total vaccination cost (incl. screening)** | $25,078,064 | $95,990,655 | $139,472,616 | $235,353,689 | $334,437,563 |
| **Vaccination cost per child** | | | | | $71.14 |
| **Total required cost** | $25,452,236 | $96,899,358 | $140,604,040 | $236,431,661 | $335,524,443 |
| **Required cost per child** | | | | | $72.11 |
| **Cost per QALY gained (healthcare)** | | | | | $5,733 |
| **Cost per QALY gained (payer)** | | | | | $5,791 |
| **Cost per life saved (healthcare)** | | | | | $335,696 |
| **Cost per life saved (payer)** | | | | | $339,071 |
| **Healthcare budget** | $4,502,303,273 | $4,566,483,106 | $4,631,577,814 | $4,697,600,441 | $4,764,564,213 |
| **Routine immunization budget** | $292,515,137 | $307,072,148 | $323,453,288 | $339,186,019 | $355,260,353.12 |
| **Required cost for dengue vaccination** | $25,452,236 | $96,899,358 | $140,604,040 | $236,431,661 | $335,524,443 |
| **Dengue vaccination cost, compared with healthcare budget (%)** | 0.57% | 2.12% | 3.04% | 5.03% | 7.04% |

*(Continued)*

**Table 2.** (Continued)

| Year | 2020 | 2021 | 2022 | 2023 | 2024 |
|---|---|---|---|---|---|
| Dengue vaccination cost, compared with routine immunization budget (%) | 8.70% | 31.56% | 43.47% | 69.71% | 94.44% |

$5,791 per QALY-gained from the healthcare and payer perspective, respectively, which clearly confirmed that dengue vaccination would be cost-effective from both perspectives, according to the cost-effectiveness threshold of GDP per capita. This study is linear with other economic evaluation studies of new vaccines in Indonesia, which mentioned that new vaccines could be cost-effective, highly cost-effective or even cost-saving [37–39]. Despite the fact that the cost-effectiveness value from healthcare perspective is lower than payer perspective, there is no significant difference on the ICERs since the dominant role of vaccine efficacy might lead the small difference between the ICERs from both perspectives.

Our finding that dengue vaccine would be cost-effective to be introduced in a country with high level of endemicity strengthened the results from several previous studies that specifically focused in endemic countries [11, 40]. Several factors tend to make dengue vaccination particularly favourable in an endemic country, such as high incidence of dengue, high vaccination impact, and high cost per case. As a country with high incidence of dengue, the seroprevalence rate tends to be high that might lead into high vaccination impact [41]. High cost per case is associated with a high per capita GDP, as the medical cost is roughly proportional to GDP [42]. To optimize the cost-effectiveness value of dengue vaccination, those three factors should be taken into account before the nationwide vaccination will be implemented in the future. The results of sensitivity analysis in this study also reconfirmed the results from several previous studies that probability of DENV infection [25], vaccine efficacy [43], under-reporting factor [9], vaccine price [44], case fatality rate [45] and screening cost [18] are the most influential parameters affecting cost-effectiveness value of dengue vaccination.

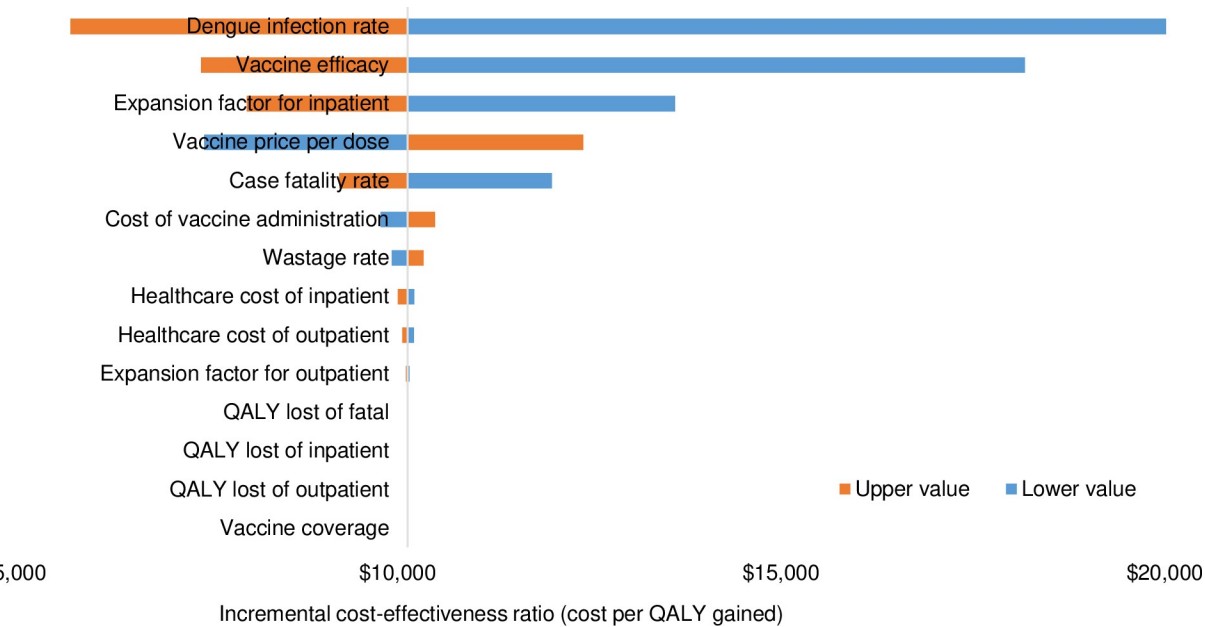

**Fig 4. One-way sensitivity analysis.**

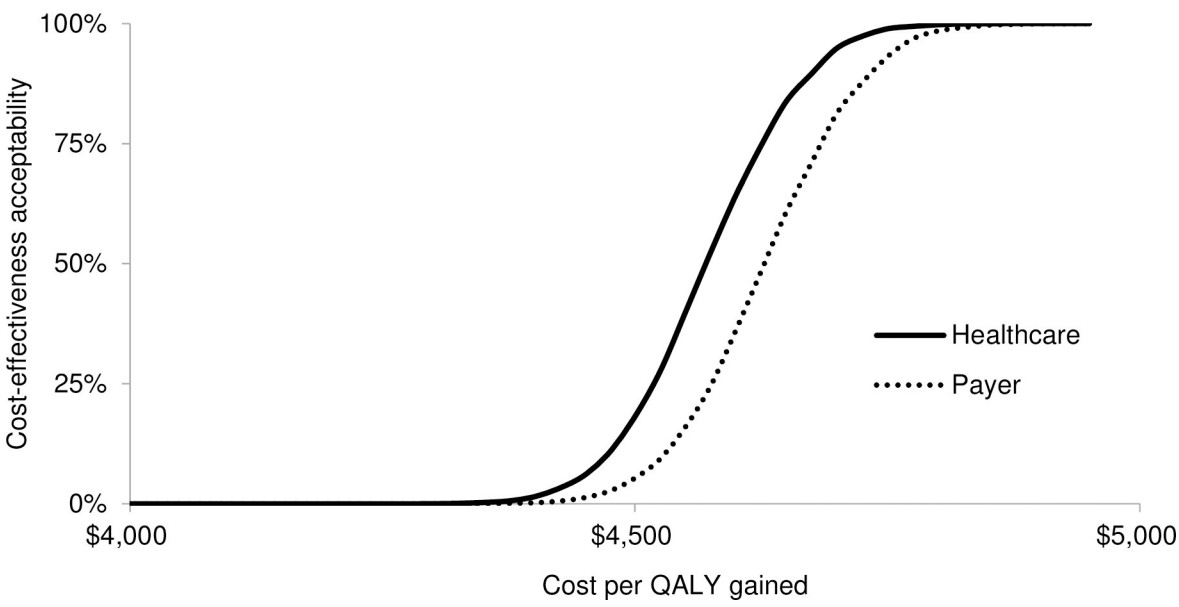

**Fig 5. Cost-effectiveness acceptability curves from the healthcare and payer perspective.**

This study is not the first economic evaluation study on dengue vaccination in Indonesia. Nevertheless, it has several major novelties. Compared to a previous study that analysed the cost-effectiveness of dengue vaccination in Indonesia as a part of ten endemic countries [11], our study has some significant differences in the process of analysis. Firstly, we focused our study specifically in Indonesia by developing a hypothetical model and taking country specific data into account. All input parameters were derived from local data, except the vaccine efficacy data. However, the key challenge to conduct economic evaluation studies in low- and middle-income countries is the difficulty in obtaining local data [46]. Secondly, we took pre-vaccination screening into account, as recommended by the WHO to minimize the risk of vaccination to

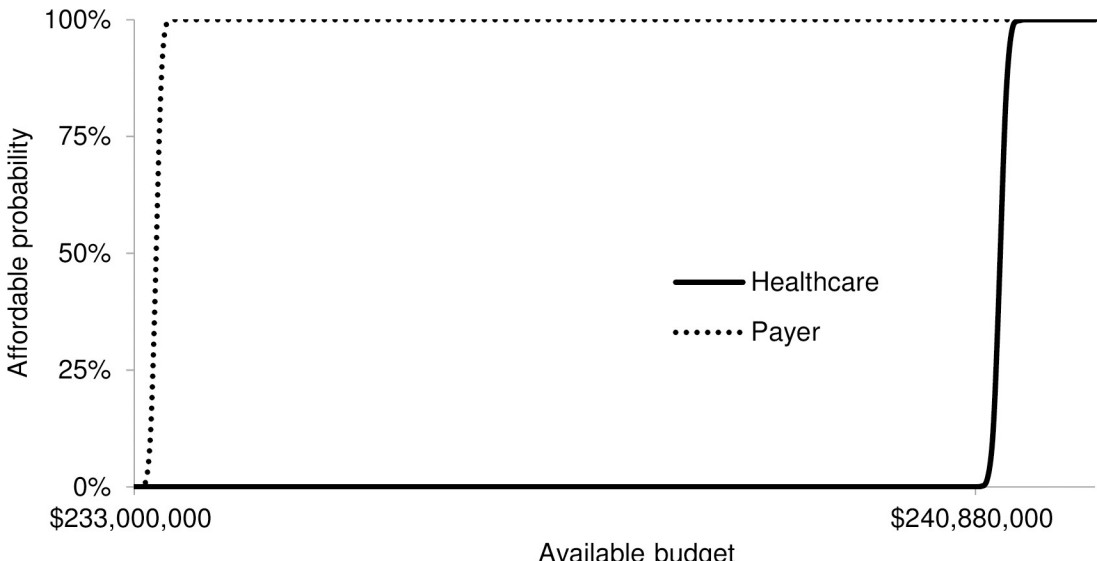

**Fig 6. Affordability curves from the healthcare and payer perspective.**

seronegative population. Thirdly, we compared two perspectives in our study: the healthcare and payer perspective. The healthcare perspective is relevant for assisting decision makers in the health sector since it considers only direct medical cost. While, the payer perspective that considers all cost covered by the Indonesian National Health Insurance System/BPJS Kesehatan, is useful to be applied because Indonesia has started to implement National Health Security since 2014. This issue would be crucial as vaccination programs have not yet been included in the benefit package of national health insurance system. Fourthly, we performed a hypothetical model by developing a stepwise on the introduction of dengue vaccine in 2020–2024. An epidemiological approach was applied since we considered the incidence rate of DENV infection in all provinces in the last 10 years to expand the introduction area. However, annual incidence of disease is one of important criteria for the prioritization of public health intervention [47]. At the end, this study also analysed the budget impact of dengue vaccination by exploring affordable required budget and making comparison with routine immunization and health expenditure budget.

Nevertheless, several limitations were found in this study. The first and main limitation is the use of the static model instead of the dynamic model due to the lack of local data on herd effect. If we took herd effect into account, the cost-effectiveness value would be more favourable. The second limitation is the lack of country specific data on the vaccine efficacy. This data was applied from a meta-analysis by using the random-effects model [22]. However, in the evidence hierarchy, a well-designed meta-analysis is at the top of the pyramid [48]. To deal with this limitation, we take this issue into account in the sensitivity analyses.

This study provides information for policy makers in Indonesia to develop a comprehensive step on including dengue vaccination into the national immunization program. To implement a nationwide dengue vaccination program with pre-vaccination screening, the government of Indonesia would require budget at $335.52 million ($72.11 per child). Compared with total national healthcare budget and routine immunization budget, the required cost for dengue vaccination would be 7.04% and 94.44%, respectively. As a country with limited healthcare and immunization budget, this situation would be very challenging to be sustainably implemented since more new vaccines are coming in the future. However, creating new fiscal space to finance new vaccination programs is very important to ensure the sustainability of such new additional programs so that they would be financed over the medium and long term and in a way that would not endanger the sustainability of the Indonesian government's financial position. New fiscal space for dengue vaccination could be created from efficiency gains in other health interventions, other vaccination programs and from dengue vaccination program itself. Expanding fiscal space could also be derived through new government financing from new revenue sources or from increased revenues, such as through economic growth, new tax administration and strengthened macroeconomic policies [36]. Hopefully, this study would assist the Indonesian government in making regulation to reduce DENV infection in Indonesia, which is in line with WHO's goal on the implementation of universal vaccination [31].

## Conclusion

Despite the fact that dengue vaccination would be cost-effective in Indonesia according to the cost-effectiveness threshold of GDP per capita, it seems unaffordable to be implemented since the total required cost for nationwide vaccination and pre-vaccination screening would be 94.44% of routine immunization budget.

## Supporting information

**S1 Appendix. Life table and cohort projection.**
(PDF)

**S2 Appendix. Estimated cases (nationwide vaccination).**
(PDF)

**S3 Appendix. Healthcare cost (nationwide vaccination).**
(PDF)

**S4 Appendix. Payer cost (nationwide vaccination).**
(PDF)

**S5 Appendix. Vaccination cost (nationwide vaccination).**
(PDF)

**S6 Appendix. QALY losses (nationwide vaccination).**
(PDF)

**S7 Appendix. Incremental cost-effectiveness ratio (nationwide vaccination).**
(PDF)

## Author Contributions

**Conceptualization:** Auliya Abdurrohim Suwantika, Mohammad Ali.

**Data curation:** Woro Supadmi.

**Formal analysis:** Auliya Abdurrohim Suwantika, Woro Supadmi.

**Funding acquisition:** Rizky Abdulah.

**Investigation:** Auliya Abdurrohim Suwantika, Woro Supadmi, Rizky Abdulah.

**Methodology:** Auliya Abdurrohim Suwantika.

**Project administration:** Rizky Abdulah.

**Resources:** Rizky Abdulah.

**Software:** Auliya Abdurrohim Suwantika, Woro Supadmi.

**Supervision:** Mohammad Ali, Rizky Abdulah.

**Validation:** Auliya Abdurrohim Suwantika, Woro Supadmi.

**Writing – original draft:** Auliya Abdurrohim Suwantika, Woro Supadmi.

**Writing – review & editing:** Mohammad Ali, Rizky Abdulah.

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
