## [Decision Letter · Decision Letter 0]

1 Jul 2020

Dear Dr. Suwantika,

Thank you very much for submitting your manuscript "Cost-Effectiveness and Budget Impact Analyses of Dengue Vaccination in Indonesia" for consideration at PLOS Neglected Tropical Diseases. As with all papers reviewed by the journal, your manuscript was reviewed by members of the editorial board and by several independent reviewers. In light of the reviews (below this email), we would like to invite the resubmission of a significantly-revised version that takes into account the reviewers' comments. 

We cannot make any decision about publication until we have seen the revised manuscript and your response to the reviewers' comments. Your revised manuscript is also likely to be sent to reviewers for further evaluation.

Sincerely,

Brett M. Forshey

Associate Editor

Paulo Pimenta

Deputy Editor

Reviewer's Responses to Questions

**Key Review Criteria Required for Acceptance?**

**Methods**

-Are the objectives of the study clearly articulated with a clear testable hypothesis stated?

-Is the study design appropriate to address the stated objectives?

-Is the population clearly described and appropriate for the hypothesis being tested?

-Is the sample size sufficient to ensure adequate power to address the hypothesis being tested?

-Were correct statistical analysis used to support conclusions?

-Are there concerns about ethical or regulatory requirements being met?

Reviewer #1: This is a very relevant study, with methods seeming to have incorporated all relevant aspects of dengue epidemiology and costing. I would like to urge the authors to be bit more extensive on the accuracy and validity of the QALY estimates. It could be considered bit more in the Discussion. I guess there is large uncertainty around them...

Reviewer #2: Methods were clearly described. Model and parameters used in the model were clearly described.

1)I would like to know why side-effects of the CYD-TDV vaccine not included in the modeling or in the scenario analysis. 2)Pre-vaccination screening methods are now recommended by the WHO for dengue vaccine introduction. I would like to see this added to the CEA and BIA, as pre-vaccination screening would be an additional programmatic cost, and this might shift CEA results to not cost-effective

Reviewer #3: As I explain in the overall comments, I think the methods are insufficient for this study.

**Results**

-Does the analysis presented match the analysis plan?

-Are the results clearly and completely presented?

-Are the figures (Tables, Images) of sufficient quality for clarity?

Reviewer #1: Please discuss the potentials of vaccination for other areas than the currently selected regions in Indonesia...

Reviewer #2: I'm concerned withe the use of the so-called WHO criteria for cost-effectiveness threshold based on a countries GDP. The WHO has stated several years ago that GDP per capita thresholds are not recommended. Also, thresholds from 1-3 GDP per capita are very easily achievable. Please suggest an alternative for Indonesia that makes more sense.

Reviewer #3: The results are well presented and match their analysis plan. They are clearly presented. Figures are clear, except for figure 3, where the bars are below the text.

**Conclusions**

-Are the conclusions supported by the data presented?

-Are the limitations of analysis clearly described?

-Do the authors discuss how these data can be helpful to advance our understanding of the topic under study?

-Is public health relevance addressed?

Reviewer #1: Adequately based on the Methods and Results

Reviewer #2: Discussion is good. Strength and weakness of the study detailed.

Reviewer #3: The conclusions are well written and show the public relevance of their study. However, the conclusions are limited by the methodology used to obtain the results (as stated in my overall comments)

**Summary and General Comments**

Reviewer #1: (No Response)

Reviewer #2: (No Response)

Reviewer #3: The manuscript describes a cost-effectiveness analysis of dengue vaccination in Indonesia using a decision tree model. The vaccine being evaluated is the CYD-TDV. The authors focused on a cohort of 9-year-old children followed for 10 years, and assumed implementation of vaccination in 6 provinces every year. Burden of disease was evaluated with a static model of incidence of dengue fever and severe dengue, as well as annual deaths. The cost effectiveness was determined using GDPs thresholds for the Incremental Cost Effectiveness Ratio by QALYs gained. The authors found that vaccination would be cost-effective but not affordable for Indonesia. 

The manuscript addresses a very important public health issue, which is to determine the cost-effectiveness of dengue vaccination in an endemic country such as Indonesia. The economic analysis seems sound and the authors presented in a clear manner. However, the methodology used by the authors ignored very important characteristics of dengue transmission and the CYD-TDV vaccine, which should not be ignored in a cost-effectiveness analysis of the implementation of this vaccine. 

1. The authors completely omitted any reference to the results from the CYD-TDV trial (e.g., Sridhar et al, 2018) and the safety issues found with the vaccine. The CYD-TDV vaccine was found to elevate the risk of severe disease in post-vaccination infections in vaccinees without prior exposure to dengue virus at the time of vaccination. This is a key characteristic of the vaccine and lead to a revision of the WHO recommendation on the use of the vaccine. The WHO now recommends that pre-vaccination screening strategy to vaccinate only seropositive individuals is the preferred option for dengue vaccine implementation (https://www.who.int/immunization/research/development/dengue_q_and_a/en/). Given that the authors do not include this elevated risk in individuals without prior exposure to dengue virus, I think their analysis is incomplete and might overestimate the economic benefits of vaccination, and more importantly they would overestimate the reduction on dengue burden with the vaccine. Many researchers have addressed this issue in cost-effectiveness analyses (Some of them are Flasche et al., 2016, Coudeville et al., 2019, España et al., 2019). Even the reference that the authors cited (Zeng et al., 2018) considers this issue with the vaccine. 

2. Dengue burden. The authors acknowledge that the static model might not be enough to reproduce the dynamics of transmission of dengue virus in Indonesia. In addition to that, I also think the model might not be sufficient because it does not include primary, secondary, and post-secondary incidence rates. Given the characteristics of the dengue vaccine, these rates are important to determine its impact on the population.

PLOS authors have the option to publish the peer review history of their article (what does this mean?). If published, this will include your full peer review and any attached files.

Reviewer #1: Yes: Maarten Jacobus Postma

Reviewer #2: Yes: Dr Tharani Loganathan

Reviewer #3: No
---

## [Decision Letter · Decision Letter 1]

16 Nov 2020

Dear Dr. Suwantika,

Thank you very much for submitting your manuscript "Cost-Effectiveness and Budget Impact Analyses of Dengue Vaccination in Indonesia" for consideration at PLOS Neglected Tropical Diseases. As with all papers reviewed by the journal, your manuscript was reviewed by members of the editorial board and by several independent reviewers. In light of the reviews (below this email), we would like to invite the resubmission of a significantly-revised version that takes into account the reviewers' comments. 

We cannot make any decision about publication until we have seen the revised manuscript and your response to the reviewers' comments. Your revised manuscript is also likely to be sent to reviewers for further evaluation.

Sincerely,

Brett M. Forshey

Associate Editor

Paulo Pimenta

Deputy Editor

The Reviewers believe that some of the issues raised in a previous round of review were addressed. However, there are still a number of other remaining concerns they raised that need to be considered. Specifically:

- Please address how the GDP-based thresholds used in this analysis line up with WHO recommendations and adjust text accordingly (as recommended by Reviewer 2)

- Provide better description of the screening assay and better data points and reference for sensitivity and specificity of the screening assay - the reference and sens/spec provided are for a classifier for acute dengue, not for measuring past exposure (which I believe would be an IgG ELISA)

- Also, please review whether the calculation for adjusting the seroprevalence is correct - I'm not sure I follow the rationale for the PE9*sensitivity + (1-PE9)(1-specificity) calculation. Please review whether this is correct (vs perhaps PE9*specificity + (1-PE9)(1-sensitivity)) or even necessary. Won't the screening assay have some level of false positives and false negatives that need to be considered, rather than adjusting the estimated population prevalence up front?

- Along those lines, as one of the Reviewers mentioned, it's probably important to address potential impact of vaccination on individuals with no prior dengue exposure (eg false positive on the screen test) in terms of increasing potential for severe disease

- Provide a clearer explanation of how the 10 year follow-up fits in with the budget analysis

There are a number of other points to consider from the Reviewers below. In addition, there are a few other areas that need to be clarify:

- line 90: Please provide a reference for the program @Risk

- lines 93-96: Please explain how these DF, DHF, and DSS probabilities were derived, beyond "national administrative data and the results of several previous studies" - how were all of these data sources used and merged together into those values?

- line 225: The sentence states that vaccination could avert 84,155 cases annually, but I don't see where that calculation comes from

- line 254: The sentence seems to be incomplete: "...the difficulty in obtaining." - obtaining what?

- line 270: "The static model tends to over-estimate the cost-effectiveness results due to the inability to incorporate the herd effect. In particular, if we took herd effect into account, there would be an even more favorable cost-effectiveness." I'm not sure I follow - sounds like the model would overestimate cost-effectiveness, but then you are saying that the change would be more favorable?

Reviewer's Responses to Questions

**Key Review Criteria Required for Acceptance?**

**Methods**

-Are the objectives of the study clearly articulated with a clear testable hypothesis stated?

-Is the study design appropriate to address the stated objectives?

-Is the population clearly described and appropriate for the hypothesis being tested?

-Is the sample size sufficient to ensure adequate power to address the hypothesis being tested?

-Were correct statistical analysis used to support conclusions?

-Are there concerns about ethical or regulatory requirements being met?

Reviewer #2: 1. Side-effects of CYD-TDV vaccine – response acceptable

2. Prevaccination screening methods- now added in the CEA. I would like this to be explained in the Introduction. It is important to state that this vaccine has safety concerns among immunologically naïve, as such WHO has recommended prevaccination screening before immunisation.

3. WHO criteria for cost-effectiveness. WHO has denied recommending thresholds based on GDP for cost-effectiveness. 

1. Marseille E, Larson B, Kazi DS, et al. Thresholds for the cost–effectiveness of interventions: alternative approaches. Bulletin of the World Health Organization 2015;93(2):118-24.

2. Bertram MY, Lauer JA, De Joncheere K, et al. Cost-effectiveness thresholds: pros and cons. Bulletin of the World Health Organization 2016;94(12):925-30.

3. 1. Eichler H-G, Kong SX, Gerth WC, et al. Use of cost-effectiveness analysis in health-care resource allocation decision-making: How are cost-effectiveness thresholds expected to emerge? Value in Health 2004;7(5):518-28.

4. Newall A, Jit M, Hutubessy R. Are current cost-effectiveness thresholds for low-and middle-income countries useful? Examples from the world of vaccines. Pharmacoeconomics 2014;32(6):525-31.

5. Loganathan T, Ng C-W, Lee W-S, et al. Thresholds for decision-making: informing the cost-effectiveness and affordability of rotavirus vaccines in Malaysia. Health Policy and Planning 2018;33(2):204-14. doi: 10.1093/heapol/czx166

Please remove references to the WHO threshold from the manuscript, as they are misleading. Please provide a better justification for using GDP based thresholds.

Page 10, Line 160-163:

The incremental cost-effectiveness ratio (ICER) was evaluated by using the WHO’s criteria on cost-effectiveness of universal immunization according to the GDP per capita, (i) highly cost-effective (less than one GDP per capita); (ii) cost-effective (between 1-3 times GDP per capita); and (iii) cost-ineffective (more than 3 times GDP per capita) [26].

Reviewer #3: (No Response)

**Results**

-Does the analysis presented match the analysis plan?

-Are the results clearly and completely presented?

-Are the figures (Tables, Images) of sufficient quality for clarity?

Reviewer #2: (No Response)

Reviewer #3: (No Response)

**Conclusions**

-Are the conclusions supported by the data presented?

-Are the limitations of analysis clearly described?

-Do the authors discuss how these data can be helpful to advance our understanding of the topic under study?

-Is public health relevance addressed?

Reviewer #2: (No Response)

Reviewer #3: (No Response)

**Editorial and Data Presentation Modifications?**

Reviewer #2: (No Response)

Reviewer #3: (No Response)

**Summary and General Comments**

Reviewer #2: (No Response)

Reviewer #3: The revision of this manuscript attempted to address my main concerns from the first version. I appreciate the authors' efforts. However, there are relevant points that I consider need to be addressed. My main concern is the implementation of the pre-vaccination screening strategy.

 1. Sensitivity and specificity of the tests used to determine previous exposure to DENV. The authors describe SP9 as the seropositivity in 9-year-olds, i.e., the proportion of children who would be eligible for vaccination. However, the authors used sensitivity and specificity values of dengue diagnosis, which only detects active DENV infection. Sensitivity and specificity values for pre-vaccination screening should refer to tests to determine previous infection.

 2. The model omits the elevated risk of severe dengue on vaccinated individuals without previous exposure to dengue virus. Specifically, the decision tree should allow for screening and vaccination of truly seropositive individuals (sensitivity * PE) and truly negative individuals ((1 - specificity) * (1 - PE)).

 3. The results about disease rates and cost-effectiveness should include the elevated rates of hospitalization of individuals without previous exposure to DENV.

 4. The authors should discuss the characteristics of the vaccine (CYD-TDV) and its implications for implementation. Only the Philippines and Brazil have attempted to include the vaccine in their vaccination programs. However, due to the negative effects of CYD-TDV, those vaccination strategies were halted. In the manuscript, it's unclear the reason for a pre-vaccination screening strategy, which is the elevated risk in individuals without previous exposure to DENV. 

Other concerns:

 - Line 49,52: "dengue infection" should be "dengue virus infection". In general, I would recommend the authors to double check their use of dengue and dengue virus. Dengue refers to the disease, and dengue virus (DENV) to the virus. 

 - Lines 64-65: "dengue vaccination has been confirmed to be a cost-effective strategy for dengue control". I don't understand that claim. Dengue vaccine was not available in 1993, how can it be confirmed to be cost-effective? Also, given the unexpected elevated risk on seronegative individuals from CYD-TDV, I would recommend to only reference recent studies that take into account the specific characteristics of CYD-TDV for any claims on cost-effectiveness.

 - Line 65: "it has been confirmed". Similar to my previous comment. I disagree, it's expected to be cost-effective, but it's not confirmed.

 - Line 68-69: "incidence and mortality rates due to dengue infection in Indonesia are relatively high." Please add a reference and numbers to support that statement.

 - Lines 68-73: This paragraph speculates the reasons to not implement the vaccine, but it ignores the situation in other countries. To my knowledge, the vaccine has not been implemented in any other country after it was retracted from the Philippines. Other than the economic constraints of Indonesia to implement the vaccine, are there any other reasons that other countries, and Indonesia, have not implemented the vaccine?

 - The authors should elaborate on the characteristics of CYD-TDV and why the WHO recommends pre-vaccination screening strategies.

 - It's not clear to me why the budget analysis is from 2020-2024 whereas the cohort is followed by 10 years

 - Line 82: What's the incidence rate in the last 10 years?

 - Lines 96-100: I don't understand how the authors estimated these probabilities.

 - In general, the clinical manifestations of dengue are grouped into Dengue Fever and Sever Dengue (https://www.ncbi.nlm.nih.gov/pmc/articles/PMC3202316/). It's unclear why the authors decide to use the classification of Dengue Hemorrhagic Fever and Dengue Shock Syndrome.

 - Line 120: The sensitivity and specificity should be to detect previous exposure to DENV not current infection. There should also be a sensitivity analysis over these values.

 - Line 123 - 131: I'm not sure the authors addressed my concerns on the elevated risk for vaccinated seronegative individuals. As it's presented in line 117 (From reference 16), there is a proportion of children without prior exposure to dengue virus who are vaccinated, i.e., (1-PE9)x(1-specificity). These children would have an elevated risk of experiencing severe dengue, which should be included in the costs of the vaccination program. I do not see that estimate in the manuscript.

 - Would like to split the results in terms of cases averted into: cases averted by vaccination of previously exposed and the additional cases caused by vaccination of non-previously exposed individuals.

 - Where did the authors obtained the $10 cost for pre-vaccination screening?

 - QALYs calculation should include the cost of of additional hospitalizations caused by vaccination of non-exposed individuals

PLOS authors have the option to publish the peer review history of their article (what does this mean?). If published, this will include your full peer review and any attached files.

Reviewer #2: Yes: Dr Tharani Loganathan

Reviewer #3: No
---

## [Decision Letter · Decision Letter 2]

20 Jul 2021

Dear Dr. Suwantika,

We are pleased to inform you that your manuscript 'Cost-Effectiveness and Budget Impact Analyses of Dengue Vaccination in Indonesia' has been provisionally accepted for publication in PLOS Neglected Tropical Diseases.

Best regards,

Brett M. Forshey

Associate Editor

Paulo Pimenta

Deputy Editor

---

## [Editor Report · Acceptance letter]

9 Aug 2021

Dear Dr. Suwantika,

We are delighted to inform you that your manuscript, "Cost-Effectiveness and Budget Impact Analyses of Dengue Vaccination in Indonesia," has been formally accepted for publication in PLOS Neglected Tropical Diseases.

Best regards,

Shaden Kamhawi

co-Editor-in-Chief

Paul Brindley

co-Editor-in-Chief
